# Canine Seventh Lumbar Vertebra Fracture: A Systematic Review

**DOI:** 10.3390/ani12020193

**Published:** 2022-01-13

**Authors:** Chiara Caterino, Federica Aragosa, Giovanni Della Valle, Gerardo Fatone

**Affiliations:** Department of Veterinary Medicine and Animal Production, University of Naples “Federico II”, 80137 Naples, Italy; chiara.caterino@unina.it (C.C.); federica.aragosa@unina.it (F.A.); fatone@unina.it (G.F.)

**Keywords:** spinal fracture, L7 fracture, traumatic lumbosacral joint dislocation, spine stabilization, dog

## Abstract

**Simple Summary:**

The aim of this paper is to review the available literature about canine seventh vertebra lumbar fracture. According to PRISMA guidelines, the authors detected data about the clinical presentation, surgical techniques, and the outcome of this rare traumatic injury that is poorly reported and described. The major problem addressed was the lack of prospective studies with a large study population and univocal data collection about complications, outcome, and post-surgical treatment.

**Abstract:**

(1) Background: Lumbosacral traumatic injuries are reported as 39% of canine vertebral lesions. This area is prone to fracture and luxation. Several surgical techniques were described from 1975 to 2021 to stabilize the traumatic injuries of the lumbosacral junction. This report aims to critically review the available literature focused on clinical presentation, surgical techniques, and follow-up of the lumbar vertebra fracture. (2) Methods: Three bibliographic databases: PubMed, Google Scholar, and Scopus were used with a board search of Lumbosacral junction fracture AND, of L7 fracture AND (canine OR dog). The Joanna Briggs Institute (JBI) Critical Appraisal Checklist for case reports and case series was applied for the studies included. (3) Results: A total of 432 reports yielded only nine that met the inclusion criteria. Non-ambulatory paraparesis/plegia, sciatic nerve involvement, faecal/urinary incontinence, and severe back lumbar pain were the most reported signs. Survey radiographs were the most reported technique to confirm the diagnoses. The surgical treatment was reported in all reports examined with a good long-term prognosis. (4) Conclusions: The seventh lumbar vertebra fracture, despite the different surgical techniques performed, had a favourable prognosis for long-term outcome and neurological recovery.

## 1. Introduction

Spinal fracture/luxation in small animals is relatively common and usually associated with a road car injury and falls from heights [1]. Other causes are represented by animal attacks, gunshot wounds, and nontraumatic causes, such as neoplasia, infection, and metabolic disease [2]. The lumbar tract represents the second most common region of the spine affected by fractures and luxation after the thoracolumbar region [3]. 

The lumbosacral area is considered a high mobile functional unit, in which the stress concentration at the junction of relatively mobile and not the mobile spinal tract, makes this area prone to joint displacement accounting the 39% of all vertebral lesions [4]. Lumbosacral traumatic injuries essentially are represented by the fracture of the seventh lumbar (L7) vertebra with a short oblique vertebral body fracture [5] and a traumatic lumbosacral joint dislocation (TLSJD) with a cranioventral displacement of the sacrum without the typical fracture of L7 and articular facets [6]. 

The first published report of fractures of L7 in the dog was in 1975 [7], and, in the last 46 years, different techniques have been described to stabilize these fractures [5,6,7,8,9,10,11,12,13,14]. The clinical presentation and the outcomes related to these techniques have been reported but in small case series. To the best of our knowledge, no comprehensive review is available of information on the lumbar vertebra fracture in dogs. Therefore, the aim of this review is to identify evidence about anatomy and biomechanics, clinical presentation, diagnosis, treatment, and outcome of L7 fractures by a systematic review of the available literature.

## 2. Materials and Methods

### 2.1. Aim and Literature Search Strategy

This systematic literature review followed the PRISMA (Preferred Reporting Items for Systematic reviews and Meta-Analyses) flowchart and is in accordance with PRISMA’s statement [15], including published case reports and retrospective or prospective case series of dogs undergoing surgical treatment for L7 fracture were followed by the authors. A sound description of clinical presentation (ambulatory/non-ambulatory paraparesis or paraplegia, sciatic nerve involvement, and faecal/urinary incontinence), surgical technique description, and follow-up were considered as inclusion criteria. 

To be included, a study had to use a defined clinical outcome relating to surgery, such as recovery to ambulation. The studies in which clinical presentation, surgical treatment, and outcomes were not reported were excluded. No language restrictions were applied. Four independent reviewers (CC, GDV, FA, and GF) searched the Pubmed, Google Scholar, and Scopus databases from the inception of the databases up to June 2021. Finally, we updated the database search on 15 November 2021 examining the references cited in the study reports included in the systematic review. 

Four known relevant studies [5,13,14,16] were used to identify records within databases. Search terms were also checked using the Pubmed PubReMiner word frequency analysis tool. Candidate search terms were identified by looking at words in the titles, keywords, and abstracts of those records. The search strategy was developed by one of the author (GF) and validated by testing whether it could identify the four known relevant studies. 

The electronic search phrase used was “lumbosacral junction fracture” (canine OR dog) AND “L7 fracture” (canine OR dog) AND “lumbar fracture” (canine OR dog) for all fields. Moreover, the references list of the papers selected were critically reviewed to improve the sources. No retrospective temporal limitation from June 2021 was placed on the period searched within the database for relevant publications. A flow diagram of the literature search and study selection process is presented in Figure 1.

### 2.2. Study Selection

We reviewed titles and abstracts of the first 100 records and discussed inconsistencies until consensus was obtained. Then two researchers (FA and CC) independently screened the titles and abstracts of all articles retrieved. In the case of disagreement, consensus on which articles to screen full-text was reached by discussion. If necessary, a third researcher (GF) was referred to make the final decision. Next, two researchers (FA and CC) independently screened full texts to determine eligibility. Again, all discrepancies were resolved between the authors, with the availability of a third-party adjudicator (GDV).

### 2.3. Data Extraction

Two review authors (FA and CC) independently extracted data from eligible studies. The extracted data were compared, and any discrepancy was resolved through discussion or consulting another researcher (GF). The descriptive variables extracted were the author’s name, study year, country, sample size, study period, anatomical and biomechanical considerations, clinical presentation, imaging (% of dogs undergoing RX or CT examination), surgical technique, and follow up. 

Any measure of clinical presentation, imaging, surgical techniques, outcome, and complications was eligible for inclusion. The results for the clinical presentation were reported as the number of dogs presented with ambulatory/non-ambulatory paraparesis or paraplegia, sciatic nerve involvement, and faecal/urinary incontinence. No restrictions regarding the length of follow-up or number of measurement points was considered when interpreting study findings, and the results for outcome were reported as the number of patients that regained the ability to walk.

GDV and FA independently assessed the quality of case reports and case series included using the Joanna Briggs Institute (JBI) Critical Appraisal Checklist for case reports and case series. The case report checklist was comprised of a total of eight sub-items, and the case series checklist contained 10 sub-items. In this systematic review, we verified the case reports and case series checklists to increase the accuracy of the evaluation. 

It was evaluated as “Yes” if it was clearly described, “No” if it was not presented, “Unclear,” if it was not clear, and “Not applicable” if it could not be applied. All disagreements were resolved by discussion. In each of the sub-items, the number of studies evaluated as “Yes”, “No”, “Unclear”, and “Not applicable” were reported. According to Munn et al. (2019) guidance [17], we presented the results of critical appraisal for all questions via a table rather than summarizing with a score.

Considering the inclusion of different study designs and interventions, the Synthesis Without Meta-analysis (SwiM) checklist [18] was applied. Meta-analyses could not be undertaken due to the heterogeneity of surgeries and study designs.

## 3. Results

### 3.1. Database Review

We found 466 records in the database searching from 1975 to June 2021. After the removal of duplicates, we screened 432 records, from which we reviewed 28 full-text documents, and finally included eight papers [5,10,11,12,13,16,19,20]. Later, we searched records from the reference lists of initially included studies, and found one article that fulfilled our inclusion criteria [19]. 

Only nine papers met the inclusion criteria, and they reported L7 fractures with a median of 6.3 cases (range 1–17). We excluded 14 studies from our review due to treating other vertebral fractures than L7 (*n* = 9), generic study on vertebral fractures (*n* = 3), and biomechanical studies (*n* = 3). The included studies, year, study design, number of cases, clinical presentation, surgical technique, and outcome are summarised in Table 1.

### 3.2. Anatomical and Biomechanical Considerations

The LS joint is the junction between the last lumbar vertebra L7 and the sacrum (S1-2-3). Unlike other spine tracts, the LS area is significantly wider and stiffer and able to guarantee all the necessary support for the propulsive forces. However, the junction between the mobile lumbar spine and the rigid and immobile sacrum is a stress concentrator, and consequently one of the most critical sites in case of traumatic events [21]. According to the classification of the spinal fracture principle, the so-called “three-column spine”, the LS can be divided into dorsal, middle, and ventral compartments. 

The dorsal compartment includes the spinous process of L7, the vertebral lamina, the articular process stabilized by a robust joint capsule, the mamillary process, and the supra and interspinous ligament that act as tension band between the junctions. The middle compartment is represented by the dorsal longitudinal ligament, the floor of the vertebral canal, and the dorsal part of the annulus fibrosus. The ventral compartment includes the lateral and ventral aspect of the annulus fibrosus, the nucleus pulposus, and the ventral longitudinal ligament [22]. 

The spine stability is provided by a complex network of supporting tissues. The intervertebral disc (IVD), articular process, and vertebral body are principally involved in spine stability [21]. The IVD failure, usually due to a flexion loading force, significantly increase the lateral bending and the rotational instability. 

Flexion and rotation, acting as simultaneous loading forces, are associated with the vertebral body and articular facet fractures, as in L7 fractures resulting from a combination of compression, lateral translation, and rotational forces [3,7]. The terminal part of the spinal cord lodged within the vertebral canal is anatomically represented by the conus medullaris with its nerve roots that caudally transverse the LS and are defined as Cauda Equina. 

Generally, the spinal cord ends at the level of the L5 or L6. Although, in some small dogs and cats, the cord can end at the level of the L7 or even S1. At this level, nerve roots of the cauda equina are more resilient to trauma than the cord itself [12,23]. The neurological deficits of nerve roots inside the LS vertebral canal are basically due to traction, compression, or avulsion of the nervous fibre resulting in functional deficits of the pudendal nerve and sacral plexus [21].

### 3.3. Clinical Presentation

The lumbosacral joint fracture causes severe back lumbar pain due to mechanical instability and secondary neurological deficits. The clinical presentation may consist of an ambulatory paraparesis, non-ambulatory paraparesis, or plegia from sciatic nerve involvement and faecal/urinary incontinence. Moreover, concomitant injuries (i.e., hindlimb fractures, concomitant pelvic fractures and coxofemoral luxation, and severe soft tissues damage) are often reported and need to be addressed before the LS surgical approach. 

Ambulatory paraparesis was detected in 31.5% (12/38) of the cases, non/ambulatory paraparesis or plegia in 60.5% (23/38), sciatic nerve involvement in 21.0% (8/38), and faecal/urinary incontinence in 42.1% (16/38). Moreover, at the rump inspection, different authors reported a typical dorsal displacement of the spinous process of L7 as compared to the level of the ilial wing in L7 fracture [5,6,8,11,12,13,16,19,20]. Decreased or absent pelvic limb withdrawal reflex secondary to sciatic nerve dysfunction as well as faecal and urinary incontinence due to damage of the pudendal nerve and sacral plexus were detected and reported by several authors. As reported in the literature, all dogs presented with L7 fracture had neurological signs consistent with an L6-S2 myelopathy [5,6].

### 3.4. Imaging

According to the reviewed literature, survey radiographs were the most common imaging method used [5,6,8,11,12,13,16,19,20]. All patients with L7 fractures underwent survey radiographs in latero-lateral (LL) recumbency to confirm the diagnoses and assess the fracture or luxation [5,6,8,11,12,13,16,19,20] (Figure 2). The ventro-dorsal (VD) view was not performed for L7 fracture. In 4/38 patients, Computer Tomography (CT)-myelography was used [16]. No reviewed report referred to advanced imaging modalities, such as MRI.

### 3.5. Treatment

According to the reviewed literature, none of the patients with L7 fractures underwent to conservative treatment. Among the reports considered, several surgical techniques were performed to reduce and stabilize the L7 fracture. Dulisch and colleagues, in 1981, used a double transilial pins and plastic plates to treat the L7 fracture in one patient [19]. McAnulty and colleagues in 1986 described the use of the Steinmann pins placed transversely through both ilial wings at the level of the sacral dorsal lamina, bent at a right angle, and then placed alongside the laminae and attached to the articular facets and spinous process by a stainless steel wire [8]. 

In 1993, Ullman and Boudrieau reported the use of two crossed transilial pins secured with double Kirschner clamps placed adjacent to the ilial wings under the muscular plane [10]. Beaver and colleagues in 1996 reported the use of screws bilaterally placed into the ilial wing and into the bodies of sixth and fifth lumbar vertebra incorporated into Polymethyl-methacrylate (PMMA) [12]. Harrington and Bagley, in 1998 in two dogs with L7 fractures, used Steinmann pins inserted across the L7-S1 articular facets and a different number of pins placed as anchors and embedded in PMMA [11]. 

In 2007, Weh and colleagues reported the use of four positive-profile threaded pins into a L7-L6 pedicle body embedded in PMMA [13]. Wheeler and colleagues in 2007 described the use of trans-sacral and transilial pins [20]. Di Dona and colleagues in 2016 reports the use of double transilial pins externally fixed with two Kirschner clamps in 17 patients [5]. Segal and colleagues in 2018 addressed the L7 fracture in six dogs by using bilateral string-of-pearls (SOP) fixed to the lateral aspect of the vertebral body cranial to L7 and ilio-sacral joints [16].

### 3.6. Complications and Prognosis

According to the literature reviewed, all patients surgically treated regain the ability to deambulate [5,6,8,11,12,13,16,19,20]. The failure of fixation is the primary complication detected, and incorrect choice of fixation technique, incorrect execution of fixation technique, and incorrect post-operative management are the factors leading to instability. Bone lysis around pins and pin tract infection due to inflammation/infection and breakage of pins/screw were reported as the main causes of implant failure [8,13,20]. 

However, the long-term prognosis can be considered favourable for outcome and neurological recovery. Caudal lumbar lesions had a better neurological status as compared to more cranially located vertebral injuries [12]. Moreover, the bone healing via callus of L7 body vertebra quickly stabilizes the fractured site ensuring a rapid solution of the orthopaedic injuries and a good neurological recovery [6].

## 4. Discussion

This paper reviewed and summarized the available evidence of the seventh lumbar vertebra fracture in dogs, through a systematic review of the literature. Although the lumbosacral traumatic injuries represent 39% of all vertebral fractures [4], limited evidence was found in the veterinary literature with a sound description of clinical presentation, treatment, and follow-up. Overall, based on the findings of this review, there is a limited number of animal studies focused on this topic and data available derived by case reports and case series with a limited number of patience [5,6,8,11,12,13,16,19,20]. 

We need to exercise caution in interpreting these findings due to the small number of studies, the design, and the risk of bias. The paucity of prospective or retrospective studies with a large sample poses a major challenge for investigators. On the other hand, regarding the examined literature, the anatomical and biomechanical aspects are well defined [3,7,22]. 

The clinical presentation reflects severe back lumbar pain due to the instability of the junction. An alternate anatomical profile of the rump due to the ventral displacement of the sacrum combined with non-ambulatory paraparesis/paraplegia likely due to hyperalgesia, sciatic nerve involvement, and faecal/urinary incontinence were the most frequent signs reported. The neurological deficits encountered were related to the traction and avulsion of the pudendal nerve and sacral plexus [5,6].

The imaging technique performed to confirm the diagnoses and to address the lesion was the direct radiograph projection in latero-lateral as observed by the authors of the papers reviewed [5,6,8,11,12,13,16,19,20]. Advanced imaging modalities are preferred in human medicine but little is published for the lumbosacral region in veterinary patients [14,16]. Computer Tomography examination was reported only in one paper [16]. Although the ideal advanced imaging modality is CT or MRI, radiographs are sufficient to diagnose the majority of vertebral fractures and, in particular, a L7 fracture [5,6,8,11,12,13,16,19,20]. 

Owing to the fact that multiple fractures occur in approximately 5% to 10% of all affected animals [4], it is prudent to assess the whole vertebral column. The result of this review leads to the conclusion that radiographic evaluation should be considered sufficient for diagnosis. However further observations are needed to assess the feasibility of this technique—in particular, in the case of the use of a plate that would necessitate a more accurate definition of the bone involved.

In the surgical treatment, the most common difficulty reported was the approach to the lateral aspect of the L6 to L7 vertebral bodies, which is obscured by the wings of the ilium [16]. Not all studies agree on this finding since the presence of ilial wings represents a concern for plate (SOP) and screws or pins and PMMA placement, while external fixation in this area is easier [24].

When comparing bilateral constructs, such as pins or screws and PMMA or even plates to external fixation, the formers require a bilateral surgical approach and, therefore, bilateral skeletonization of the column. As a consequence, this leads to a prolonged surgical time and an increase in soft tissue disruption with a consequent reduction of the blood supply and, therefore, a lengthening of the healing time [24]. Comparing an SOP plate and pins and PMMA fixation, Nel et al. in 2017 postulated that the second technique requires less soft tissue dissection for placement; additionally, the use of four threaded pins bundled with PMMA, compared to screw insertion (usually 10) and plate contouring and placement, resulted in a shorter time of surgery [25]. 

On the other hand, the use of PMMA increases the potential risk of infection and of thermal injury from the exothermic reaction during the application of PMMA. Currently, the incorporation of antibiotics into the polymer powder before mixing can reduce this risk. The use of PMMA and bone screws has several advantages for vertebral stabilization; pins are easy to apply but, in comparison with bone screws, are more likely to migrate and are less resistant to pullout [12]. 

In addition, the strength of the pins and PMMA construct is strongly influenced by the diameter of the pins used; the relatively larger pedicle of L7 as compared with other vertebra allows placement into both the pedicles and the body. This trajectory (caudolateral to the caudal articular process of L7), associated with familiarity with sacroiliac anatomy, avoids interference with the iliac wing during placement, perhaps due to a more horizontal angle directed into the vertebral body only [13]. The pin insertion angle suggested in veterinary literature is about 30° to 60° from the horizontal plane [26,27]; however, the rate of complications is much lower when pins are placed under fluoroscopic guidance [26].

Finally, to maximize the strength of the fixation and minimize the change of implant failure, it is important to use a sufficient amount of PMMA, in particular in large breed dogs, because a lack could result in a fracture of the material [12] and, consequently, a second surgery. Without PMMA, the risk of implant failure is major as reported by McAnulty et al. (1986) in which a dog with an L7 fracture (case n.3) had less than anatomic reduction due to the omission of an additional bend to contour the pins to a normal lumbosacral angle [8]. 

Harrington et al. (1998) reported a formation of a seroma in concomitance to a pin migration 6 weeks post-surgery. The minor complication was resolved after the remotion of the pin, which was performed after the radiographic assessment of bone healing. In the papers analysed in this review, none of the authors reported discomfort because of implant placement [5,6,8,11,12,13,16,19,20]. Weh et al. (2007) reported a case of seroma over the implant 2 days post-operatively that was resolved in 1 week with three times daily warm compresses over the site [13]. 

Only one case of L7 fracture was treated with external skeletal fixation [20]: the application of the pins and the reduction of the fracture was achieved under a fluoroscopic guided closed reduction. The use of fluoroscopy led to the placement of the pins without complications; however, the absence of predrilling the bone before the application of the pins caused pin-bone loosening [20,28,29]. 

Minor complications associated with external fixation were skin inflammation/infection. This problem was easily resolved with appropriate antibiotics treatment and implant care by the owners [20]. Di Dona et al. (2016), through the use of percutaneous transilial pinning, demonstrated that an acceptable alignment of the vertebral canal was easily achieved. An advantage of their technique is the ease of implant removal after bone consolidation [5]. 

As described above for ESF, even with this technique, minor complications are due to pin-tract infection and pin-bone loosening [30]. This instability was addressed by moving the clamps medially and, thus, modifying the distance between the two pins bilaterally. Another important aspect is the discomfort of the patients [14,16]. Other minor complications were back pain following surgery in one dog and acute lameness after the remotion of the implant. The majority of dogs (15 out 17) returned to normal function with good-to-excellent outcomes; therefore, disregarding the minor complication, percutaneous transilial pinning can be considered as a possible treatment for dogs with L7 fractures.

## 5. Conclusions

In conclusion, the data from our review surprisingly indicate that, despite the diversity of surgical techniques applied, the prognosis is good for this type of injury. The bone healing that is achieved, via callus, results in a rapid resolution of orthopaedic injuries and a few neurological complications. Therefore, we can assume that the L7 fracture can be considered a problem of orthopaedic instability with limited long term neurological consequences. Our considerations are consistent with those reported by other authors [5,20] according to whom the neurological improvement and outcome are not related to the shortening of the involved vertebral body or residual vertebral canal compromise but is rather related to the initial neurological damage.

It is likely that the use of internal osteosynthesis systems, such as contourable plates, which allow reducing the width of the surgical access and a better adaptation to the irregular surface of the vertebrae, could optimize post-operative management and accelerate the healing processes of the fracture; however, further studies are needed.

## Figures and Tables

**Figure 1 animals-12-00193-f001:**
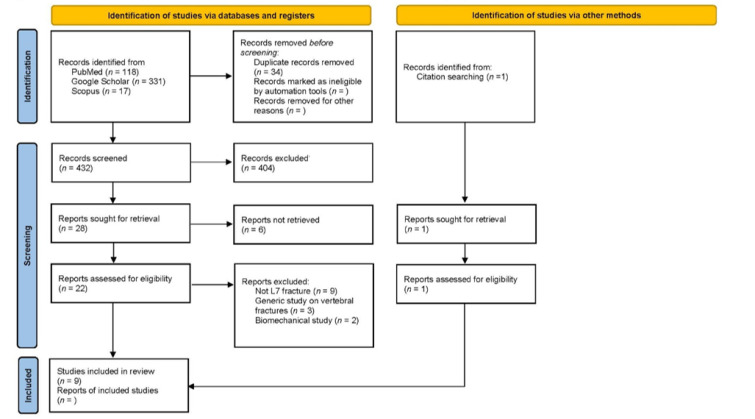
PRISMA 2020 flow diagram for new systematic reviews, which included searches of databases, registers and other sources.

**Figure 2 animals-12-00193-f002:**
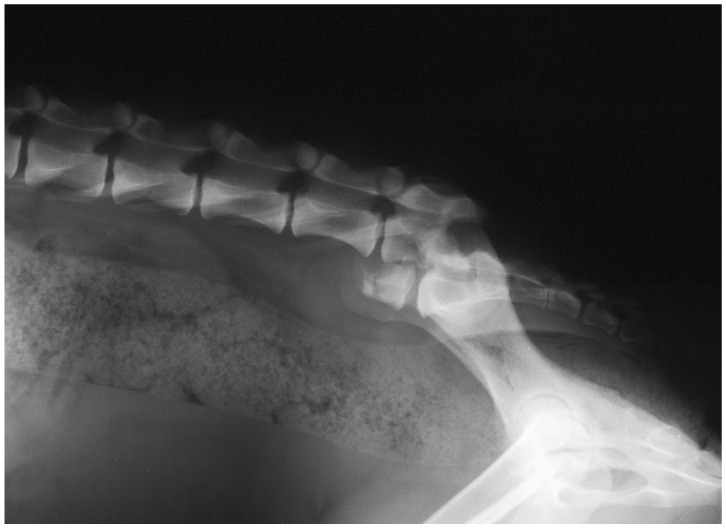
Post-trauma lateral radiographic projection of the seventh lumbar vertebra fracture showing cranio-ventral displacement of the sacrum.

**Table 1 animals-12-00193-t001:** List of reviewed papers: Type of paper, clinical presentation, surgical technique, and outcome.

Authors	Study Design	N. Cases	Paresis Ambulatory	Paresis/Plegia	Sciatic Nerve Sensitivity	Urinary/Faecal Incontinence	Post-Operative Recovery	Technique	Year
Dulish et al.	Case report	1	0	1	1	1	100%	Transilial pins and plastic plates	1981
McAnulty et al.	Case series	1	0	1	0	1	100%	Transilial pins	1986
Ullman et al.	Case series	6	6	0	0	6	100%	Double crossed pins	1993
Beaver et al.	Case series	2	0	2	2	2	100%	Screws and PMMA	1996
Harrington et al.	Case series	2	1	1	0	1	100%	Pins and PMMA	1998
Wheeler et al.	Case series	1	0	1	1	1	100%	Trans-sacral and transilial pins	2007
Weh et al.	Case series	5	1	4	1	0	100%	Pins and PMMA	2007
Di Dona et al.	Case series	17	5	12	3	6	100%	double transilial pins	2016
Segal et al.	Case series	6	2	4	3	2	100%	SOP plates	2018

## Data Availability

Not applicable.

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
