# Peer review of "Canine Seventh Lumbar Vertebra Fracture: A Systematic Review"

_animals, 2022, doi:10.3390/ani12020193_

Round 1

Reviewer 1 Report

See attached file.

Author Response

ANSWERS TO REVIWERS’COMMENTS

As corresponding, the authors (AU) thank the Reviewers for the important and valuable comments that helped to improve the quality of the manuscript. The AU have carefully considered every reviewer’s recommendation and the manuscript has been revised according to their suggestions.

Your Sincerely

Dr. Giovanni Della Valle

Reviewer #1:

In section 3.2. Anatomical and biomechanical considerations bibliography is missing. The authors do not include any books or literature regarding the anatomy and physiology. But this paragraph could be shortened

AU: Okay, the Manuscript has been accordingly modified. Please, see the text (section 3.2)

Lines 125-148 lack on references. The last sentence is referring to Millers Anatomy but there is no reference to the specific chapter or pages from the book. Lines 149-151

AU: Okay, the Manuscript has been accordingly modified. Please, see the text (ref: 21)

Ref. line: 149-151

lack on references. Lines 210-227

AU: Okay, the Manuscript has been accordingly modified. Please, see the text (line: 214-230)

Lines 242-245 it is not clear what the authors describe

AU: Okay, the sentences have been rephrased accordingly your suggestion. Please, see the text (line:245-247)

Lines 250-254 in not clear what the authors describe.

AU: Okay, the sentences has been rephrased accordingly your suggestion. Please, see the text (line:252-256)

Line 265 the authors mention the suggested angle of pin insertion but there is no bibliographic reference suggesting this angle unless fluoroscopy is used.

AU: Okay, the Manuscript and references have been accordingly modified. Please, see the text (line: 267-268; Ref.: 26-27).

Line 277 -278 the Authors refer to all papers reviewed but, in the end, they add only one citation. Reference is missing

AU: Okay, the Manuscript has been accordingly modified. Please, see the text (line: 278-280)

Lines 282-284 Do they refer to human medicine? The report [External fixator: surgical technique, pinless fixator, change in procedure]. Is published in a human medicine journ

AU: Okay, the Manuscript and references have been accordingly modified. The Authors cites different sources in human and veterinary medicine that explain the importance of pre-drilling to avoid pin-loosening.

Please see the text. (line:  284-286; Ref.: 20,28,29)

Line 286 it is not clear what pin management means

AU: Okay, the Manuscript has been accordingly modified. Please, see the text (line: 293)

Reviewer 2 Report

The Authors review the available literature about the canine seventh vertebra lumbar fracture. 
The literature review is extensive and critique.
It is a very intersting systematic review, well explained, with sections presented in a balanced and coherent way. introfuction isproperly documented, and the analisys of data are properly diuscussed 
I suggest minor revisions providing some comments:
Add a lateral radiograph, demonstrating typical appearance of lumbosacral fracture. 

This Manuscript is a Systematic Review, which has as its topic the canine seventh vertebra lumbar fracture. This topic is relevant why to my knowledge no comprehensive review is available of information on seventh lumbar vertebra fracture in dogs.

The review data indicates that although there is a diversity of surgical techniques applied, the prognosis is always good. Bone healing achieved through callus results in rapid resolution of orthopedic injuries and few neurological complications. Moreover, the Authors hypothesize that the use of internal osteosynthesis systems, such as shaped plates, which allow to reduce the width of the surgical access and a better adaptation to the irregular surface of the vertebrae, can optimize postoperative management and accelerate the processes fracture healing, but further studies are needed.

In my opinion the analysis methodology is very interesting. PRISMA guideline in conducting the systematic review, including published case reports, retrospective or prospective case series of dogs undergoing surgical treatment for L7 fracture were followed by the Authors. The conclusions are consistent with the evidence and arguments presented and they address the main question posed. The references are appropriate.

Author Response

ANSWERS TO REVIWERS’COMMENTS

As corresponding, the authors (AU) thank the Reviewers for the important and valuable comments that helped to improve the quality of the manuscript. The AU have carefully considered every reviewer’s recommendation and the manuscript has been revised according to their suggestions.

Your Sincerely

Dr. Giovanni Della Valle

Reviewer 3 Report

Dear Authors

Some text corrections.

It could be interesting to add a figure with representation of the fracture in L7, and the types of fractures that can happen, pre and post trauma, etc. It will give a good ilustration to the review.

Author Response

(The authors gave the same response as above.)
